# Analytical Thermal Modeling of Metal Additive Manufacturing by Heat Sink Solution

**DOI:** 10.3390/ma12162568

**Published:** 2019-08-12

**Authors:** Jinqiang Ning, Daniel E. Sievers, Hamid Garmestani, Steven Y. Liang

**Affiliations:** 1Georgia Institute of Technology, George W. Woodruff School of Mechanical Engineering, Atlanta, GA 30332, USA; 2The Boeing Company, Huntsville, AL 35824, USA; 3Georgia Institute of Technology, School of Materials Science and Engineering, NW, Atlanta, GA 30332, USA

**Keywords:** closed-form heat sink solution, heat transfer boundary condition, analytical modeling, powder bed metal additive manufacturing

## Abstract

Metal additive manufacturing can produce geometrically complex parts with effective cost. The high thermal gradients due to the repeatedly rapid heat and solidification cause defects in the produced parts, such as cracks, porosity, undesired residual stress, and part distortion. Different techniques were employed for temperature investigation. Experimental measurement and finite element method-based numerical models are limited by the restricted accessibility and expensive computational cost, respectively. The available physics-based analytical model has promising short computational efficiency without resorting to finite element method or any iteration-based simulations. However, the heat transfer boundary condition cannot be considered without the involvement of finite element method or iteration-based simulations, which significantly reduces the computational efficiency, and thus the usefulness of the developed model. This work presents an explicit and closed-form solution, namely heat sink solution, to consider the heat transfer boundary condition. The heat sink solution was developed from the moving point heat source solution based on heat transfer of convection and radiation. The part boundary is mathematically discretized into many heats sinks due to the non-uniform temperature distribution, which causes non-uniform heat loss. The temperature profiles, thermal gradients, and temperature-affected material properties are calculated and presented. Good agreements were observed upon validation against experimental molten pool measurements.

## 1. Introduction

Powder bed metal additive manufacturing (PBMAM) can produce geometrically complex parts with effective cost. For instance, with the use of powder bed metal additive manufacturing (PBMAM) configuration, high-density laser power is employed to fully melt and fuse metal powders to build parts in a layer-by-layer manner. The high thermal gradient due to the repeated rapid heating and solidification cause defects in the produced parts, such as cracks [1], porosity [2,3], undesired residual stress [4,5], and part distortion [6]. Different techniques have been developed to monitor and control temperature conditions, namely experimental measurement, finite element method (FEM)-based numerical modeling, and physics-based analytical modeling. 

In situ temperature measurements provide real-time temperature measurements during the heating state and cooling state in different additive manufacturing (AM) processes. Embedded thermocouples have commonly been used to measure the temperature on or inside the substrate [7]. Thermal imaging cameras (infrared cameras) were used in previous work to measure the temperature on exposed surfaces, including the build and the substrate [8]. Another metallographic method was developed based on the solidification microstructure, specifically the molten pool geometry, as a post-process measurement [9]. The in-situ measurement techniques cannot measure the temperature distribution inside the build. The post-process measurement technique cannot be easily implemented without extensive experimental works, such as machining, etching, and polishing. The experimental techniques were summarized and discussed in the literature regarding the measurement capabilities and implementation issues [10,11].

To address the difficulty and inconvenience in the temperature measurement, FEM-based numerical models were developed to predict the temperature distribution in AM processes with various metal powder materials. Fu et al. developed a FEM model to predict the three-dimensional temperature distribution and the temperature gradient in selective laser melting (SLM) of Ti-6Al-4V with adopted solid bulk material properties and powder material properties [12]. Closer agreement with experimental measurement was reported in the prediction using powder material properties. Michaleris developed a FEM model to predict the three-dimensional temperature distribution in laser-assisted directed metal deposition (L-DMD) with the active/inactive element method [13]. Cao et al. developed a FEM model to predict the three-dimensional temperature distribution in electron beam-assisted directed metal deposition (EB-DMD) with a Gaussian-distributed beam profile [14]. Numerical models were also employed to predict the temperatures in AM processes for Inconel alloy, aluminum alloy, steel, etc. [15,16,17]. Recent advances have taken the powder packing effect and powder size distribution into consideration [18,19]. Li et al. and Papadakis et al. developed a FEM model using single-track or single-layer temperature conditions as the basic unit to reduce the computational cost [20,21], which simplified the complex thermal behavior between tracks and layers, and thus unavoidably reduced the prediction accuracy. Numerous FEM models were reported in the literature for temperature prediction in different AM processes with different metal materials [22,23]. Although FEM models have made considerable progress in the prediction of different AM process with easy implementation to calculate the heat transfer boundary condition, the involvement of iteration-based simulation unavoidably compromises the computational efficiency, which is still the major drawback.

Physics-based analytical models have promising short computational time without resorting to FEM or any iteration-based simulation [24,25,26,27,28,29]. Analytical models were also developed for temperature prediction in AM processes. Van Elsen et al. summarized three heat source solutions by assuming the moving heat source as a point heat source, semi-ellipsoidal heat source, and uniform heat source, respectively [30]. The assumptions of isotropic and homogeneous materials and semi-infinite medium were enforced in those solutions. The moving point heat source model was originally developed by Carslaw and Jaeger [31]. The moving point heat source solution was further developed by Ning et al. for temperature prediction of multiple tracks in an absolute coordinate [32]. Cline et al. developed another analytical model assuming a Gaussian distribution for the heat source intensity profile [33]. This solution became the moving point heat source solution by reducing the heat source beam radius to zero. Rosenthal developed a moving line heat source solution for the welding process of an infinite thin plate [34]. This solution was adopted by Tan et al. and Andrew et al. to predict the temperature distribution in L-DMD [35,36]. However, the aforementioned analytical models neglected the heat transfer boundary condition, specifically the heat loss from the part boundary, which led to unoptimized prediction accuracy. Green’s function was used for temperature prediction in a bounded medium, but the high mathematical complexity significantly reduces computational efficiency [31]. Ahsan et al. further developed Cline’s solution to calculate the heat loss from the molten pool with iterative calculations based on mass and energy balance [37]. Peyre et al. and Yang et al. developed a semi-analytical model for temperature prediction in L-DMD and SLM, respectively [38,39]. FEM models were employed to consider the heat transfer boundary condition. Therefore, an analytical solution is needed to consider the heat transfer boundary condition with high computational efficiency and high prediction accuracy [40,41].

This work presents an explicit and closed-form solution, namely heat sink solution, for heat transfer boundary conditions in analytical thermal modeling of PBMAM. The heat sink solution was developed based on the heat transfer mechanism for convection and radiation. The temperature distribution was predicted using the heat sink solution and the moving point heat source solution in PBMAM of Ti6Al4V. Different numbers of heat sinks were used in the temperature prediction to investigate the influence of heat sinks. The thermal gradient and material property variations were investigated based on the temperature prediction. The optimal number of heat sinks was determined by comparison with experimental measurements. With the same number of heat sinks, the temperature distributions were predicted and validated under various process conditions.

## 2. Materials and Methods 

In this work, a closed-form solution, namely the heat sink solution, is presented to characterize the heat loss from the part boundary due to convection and radiation in PBMAM, as illustrated in Figure 1. The red arrow and green arrows represent heat input form the laser source and heat loss from convection and radiation at the part top boundary, respectively. The heat loss from the side boundary (x-z planes and y-z planes) is assumed to be negligible because of the significantly lower temperatures than the top boundary (x-y plane). *L*, *W*, and *D* denote the molten pool length, molten pool width, and molten depth at the laser heat source location, respectively.

The heat balance equation was used as the governing equation for the derivations of the heat source solution and heat sink solution. This equation describes the change of temperatures due to the energy input from a moving volumetric heat source, and thus can be employed for the PBMAM [31]. It is expressed as the following
(1)$$\frac{\partial \rho u}{\partial t}+\frac{\partial \rho HV}{\partial x}=\;\nabla \cdot \left(k\nabla T\right)+\dot{q}$$
where *u* is internal energy, *H* is enthalpy, *ρ* is density, *k* is conductivity, $\dot{q}$ is volumetric heat source, *t* is time, *x* is the distance from the heat source, *V* is the heat source moving velocity along the x-direction, and *T* is temperature. 

The heat conduction equation was derived from the heat balance equation with V=0, and du=cdT as $c\frac{\partial \rho T}{\partial t}=\;\nabla \cdot \left(k\nabla T\right)+\dot{q}$, which can be rewritten by introducing the thermal diffusivity (κ= k/ρc) as the following
(2)$$\frac{\partial^{2}T}{\partial x^{2}}+\frac{\partial^{2}T}{\partial y^{2}}+\frac{\partial^{2}T}{\partial z^{2}}=\;\frac{1}{\kappa }\frac{\partial T}{\partial t}+\dot{q}$$
where κ is thermal diffusivity (κ=k/ρc, where k,ρ, c are thermal conductivity, density, and specific heat, respectively), and x,y,z are the distances from the heat source.

The transient-state moving point heat source solution was derived from the heat conduction equation by Carslaw and Jaeger [31]. The laser heat source was assumed as a moving point heat source for a semi-infinite medium. It is expressed as the following
(3)$$\theta_{laser}\left(x,y,z,t\right)=\;\;\frac{P\eta }{8\rho c{\left(\pi \kappa \right)}^{\frac{3}{2}}}\int_{0}^{t}\frac{\exp\left[-\frac{{\left(x-V\left(t-t^{\prime }\right)\right)}^{2}+y^{2}+z^{2}}{4\kappa \left(t-t^{\prime }\right)}\right]}{{\left(t-t^{\prime }\right)}^{\frac{3}{2}}}dt^{\prime }$$
where P is heat source power, η is absorption, t is the current time, t′ is previous time, and x, y, z are the corresponding distances from the laser source.

The transient-state moving point heat source solution can be rewritten by integrating t’ from 0 to t as the following
(4)$$\theta_{laser}\left(x,\;y,z,\;t\right)=\frac{P\eta }{2Rk\pi^{\frac{3}{2}}}\exp\left(\frac{Vx}{2\kappa }\right)\int_{\frac{R}{2\sqrt{\kappa t}}}^{\infty }\exp\left[-\xi^{2}-\left(\frac{V^{2}R^{2}}{16\kappa^{2}\xi^{2}}\right)\right]d\xi$$
where $R^{2}=\;x^{2}+y^{2}+z^{2}$ is the total distance from the laser source, ξ is a time-related integration variable, which are introduced for concise expression and easy implementation of the heat source solution.

The steady-state moving point heat source solution can be derived with infinite *t*. It is expressed as
(5)$$\theta_{laser}\left(x,y,z\right)=\frac{P\eta }{4\pi kR\left(T_{m}-T_{0}\right)}\exp\left(\frac{-V\left(R+x\right)}{2\kappa }\right)$$

The heat sink solution was derived with equivalent power for heat loss from convection and radiation, and zero moving velocity. The convection and radiation can be calculated as
(6)$$Q_{conv}=Ah\left(T-T_{0}\right)$$
(7)$$Q_{rad}=\;A\varepsilon \sigma (T^{4}-T_{0}^{4})$$
where $Q_{conv}$ and $Q_{rad}$ denote heat loss due to convection and radiation, respectively, A is the area of the heat sink, h is the convection coefficient, ε is emissivity, σ is the Stefan-Boltzmann constant, T is the temperature of the heat sink that can be estimated by the moving point heat source solution, and $T_{0}$ is room temperature.

The equivalent power for heat loss at the part boundary can be expressed as (8)$$P_{equiv}=\;Ah\left(T-T_{0}\right)\;+A\varepsilon \sigma (T^{4}-T_{0}^{4})$$

Each heat sink is a portion of the part boundary that does not move as the laser heat source. Therefore, the moving velocity of each heat sink becomes zero (V=0). The heat sink solution is expressed as
(9)$$\theta_{sink}\left(x,y,z\right)=\frac{A}{4\pi \kappa R\left(T_{m}-T_{0}\right)}\left[h\left(T-T_{0}\right)+{\varepsilon \sigma (T}^{4}-T_{0}^{4})\;\right]$$

The part boundary is mathematically discretized into many sections (heat sinks), considering the non-uniform temperature distribution at the part boundary, which causes non-uniform heat loss at the boundary.
(10)$$\theta_{loss}\left(x,y,z\right)=\overset{n}{\underset{i=0}{\sum }}\frac{A_{i}}{4\pi \kappa R\left(T_{m}-T_{0}\right)}\left[h\left(T_{i}-T_{0}\right)+\varepsilon \sigma (T_{i}^{4}-T_{0}^{4})\;\right]$$
where *n* is the number of heat sinks, which can be determined with experimental calibration to accurately calculate the heat loss at the part boundary, and *i* is the index of each heat sink. 

The final temperature solution is constructed from the superposition of heat source solution and heat sink solutions as the following
(11)$$\theta_{laser}\left(x,y,z\right)=\frac{P\eta }{4\pi kR\left(T_{m}-T_{0}\right)}\exp\left(\frac{-V\left(R+x\right)}{2\kappa }\right)-A\overset{n}{\underset{i=0}{\sum }}\frac{h\left(T_{i}-T_{0}\right)+\varepsilon \sigma (T_{i}^{4}-T_{0}^{4})\;}{4\pi \kappa R\left(T_{m}-T_{0}\right)}$$

In addition, the latent heat ($L_{f}$) is considered using the heat integration method [30], in which the temperatures are lowered because phase transformation takes place with continuous heat input.
(12)$$\begin{array}{c} T=T_{m}\;(T_{m}:\;T_{m}+L_{f}/c)\\ T=T-\;L_{f}/c\;(T>T_{m}+L_{f}/c) \end{array}$$

The presented heat sink solution provides a computationally efficient method to consider the heat transfer boundary conditions in PBMAM without resorting to FEM or any iteration-based simulations. The analytical model is constructed from the superposition of closed-form solutions, namely the heat sink solution and moving point heat source solution. The promising short computational time allows temperature prediction for larger scale parts and process-parameter planning through inverse analysis. Therefore, the developed model has improved usefulness in real applications.

## 3. Results and Discussion

In this work, the three-dimensional temperature distribution was predicted by the presented model in powder bed metal additive manufacturing (PBMAM) of Ti6Al4V. The heat sink solution was derived and employed to impose the heat transfer boundary condition without significantly compensating the computational efficiency in analytical temperature modeling. The implementation of the heat sink solution was investigated by applying different numbers of heat sinks in the temperature prediction. The temperature profile, thermal gradient, and the temperature-affected material property variation were calculated in the single-track scan. The molten pool dimensions were calculated by comparing the predicted temperature to the material melting temperature. They were used to determine the optimal number of sinks in comparison with the documented experimental values. The computational time was recorded and presented. The molten pool dimensions were then calculated by the presented model with the optimal number of heat sinks under different process conditions. Validation of the calculated molten pool dimensions was included.

The presented model was constructed from the superposition of the heat source solution and heat sink solution. The temperature rise due to the heat input from the moving laser heat source was calculated using the moving point heat source solution. The power absorptivity in the heat source solution was adopted as 0.77 [12], which is related to the laser wavelength, laser-workpiece offset distance, powder material properties, and powder-packing-related surface roughness. The temperature drop due to the heat loss from the part boundary was calculated using the heat sink solution. The part top boundary was mathematically discretized into many sinks considering the non-uniform temperature distribution at the boundary, which led to non-uniform heat loss at the boundary according to Equations (6) and (7). The number of heat sinks was determined by calibration based on the experimental measurement of molten pool dimensions in the literature [2]. The documented molten pool width and depth were measured based on the solidification microstructure using an optical microscope. The material properties and thermophysical properties of Ti6Al4V are given in Table 1. The process parameters in PBMAM of Ti6Al4V are given in Table 2. Different laser powers and scanning velocities were applied to the PBMAM. The details of the process parameters and documented molten pool dimensions are given in Table A1 in Appendix A. 

The heat sink solution was derived from the moving point heat source solution, which has a singularity issue in the heat source solution. In other words, the temperature becomes infinite in the heat source solution with zero distance (Equation (5)). Therefore, the heat sink solution has an inherent singularity issue. In order to properly implement the heat sink solution, the influence of heat sinks on the temperature prediction was fully investigated. The temperature profiles and temperature gradient were calculated in single-track scans under test 1 condition (P=100 W;V=500 mm/s). The numbers of heat sinks on the top boundary were chosen as 4 × 4, 6 × 6, 8 × 8, and 10 × 10, respectively. The area of the heat sink under each setting is identical to the total area/number of heat sinks. For example, the area of each heat sink with 4 × 4 setting is identical to the total area/16. The area of each heat sink with 8 × 8 setting is identical to the total area/64. The heat loss from the side boundary was not considered because of the significantly lower temperature compared to the top boundary near the heat source location (x = 0.8 mm, y =0.5 mm). The three-dimensional temperature profiles are illustrated in Figure 2. Figure 2a–d represents the predictions with different numbers of heat sinks, namely 4 × 4 = 16 sinks, 6 × 6 = 36 heat sinks, 8 × 8 = 64 heat sinks, and 10 × 10 = 100 sinks, with a top boundary of 1 ${\mathrm{mm}}^{2}$. The red circles represent the centers of the heat sinks. The temperature profiles were predicted using the presented model with consideration of heat loss at the top boundary (plotted as solid lines). For comparison, the temperature profiles were also calculated using the point heat source solution without consideration of heat loss at the top boundary (plotted as dashed lines). As shown in the temperature profiles on the top boundary at the x-y plane, the predicted heat-affected zones considering the heat loss (solid lines) were smaller than those predicted without considering the heat loss (dashed lines) at each temperature level. The more heat sinks, the smaller the heat affected zone, and vice versa. The employed heat sink solution can reduce the overestimation of temperature levels, and thus improve the prediction accuracy. With the heat sink solution, the complete understanding of the heat transfer mechanism in PBMAM can be implemented conveniently and with computational efficiently. Therefore, the usefulness of the developed analytical model can be significantly improved in real applications. 

The temperature gradient profiles were plotted in Figure 3 with different numbers of heat sinks under test 1 condition (P=100 W;V=500 mm/s). The large temperature gradient was observed at the near heat source location (x = 0.8 mm y = 0.5 mm) and near heat sink location (marked as red circles). The materials property variations, namely the thermal conductivity and the specific heat, were plotted in Figure 4 and Figure 5 respectively. The material property variation was caused by the temperature-dependent nature and the temperature variation. 

To determine the proper number of heat sinks, the calculated molten pool dimensions were compared to the experimental values documented in the literature [2]. The molten pool dimensions were calculated by comparing the predicted temperature to the material melting temperature, as illustrated in Figure 6. The documented values were experimentally measured based on the solidification microstructure, as illustrated in Figure 7.

As shown in Figure 8, the closest agreement was observed with 6 × 6 heat sinks under test 1 conditions. The horizontal axis 0 × 0 denotes the calculated molten pool dimensions without a heat sink. In other words, the heat loss from the part boundary was not considered. Expt. denotes the experimental molten pool dimensions. The more heat sinks, the smaller the molten pool dimensions, which is consistent with the heat-affected zone dimensions. This trend confirms the instinctive trend that the more heat loss, the smaller the heat-affected zone and molten pool, and vice versa.

The calibration data from post-process temperature measurement is more reliable and less sensitive to operation than the in-situ measurements from thermocouples and infrared cameras [10,11]. The experimental molten pool measurements were conducted at least in triplicate with negligible deviation observed under each process condition. Therefore, the use of post-process measurement of molten pool dimensions for calibration purposes is acceptable. The experimental calibration process is necessary for the presented model to avoid the compensation of computational efficiency using iterative calculation-based calibration on heat sink temperatures. The optimal heat sink settings are material-dependent because the material-dependent heat transfer coefficients of convection and radiation are used in the presented model. 

Moreover, the temperature calculations were carried out with a MATLAB (MathWorks, USA) program on a personal computer running at 2.8 GHz. The part size was 1 mm × 1 mm × 0.2 mm, with increments of 2.5 μm in all directions. The computational times with 4 × 4, 6 × 6, 8 × 8, and 10 × 10 heat sinks were 49.42 s. 94.44 s, 160.28 s, and 239.62 s, respectively. For comparison, FEM models required at over an hour for similar calculation increments and prediction accuracies [12,13,16,17]. The semi-analytical models [38,39] using coarse mesh resolution for heat transfer boundary conditions will significantly reduce the prediction accuracy for large-scale parts. The presented model does not rely on FEM or any iteration-based calculation, and thus remains unaffected.

With 6 × 6 heat sink settings, the molten pool dimensions were predicted under 8 different process conditions, as shown in Table A2 in the Appendix A. The predicted molten pool dimensions were validated by the experimental measurements. The continuity of the scan tracks was confirmed from the observation on the top view. The experimental measurement was made at least in triplicate with negligible variation observed under each process condition. Close agreement was observed, as shown in Figure 9. The deviation might be caused by the simplified point heat source solution without considering the heat source profile, the adopted absorptivity, and material properties. The associated data are given in Table A2 in the Appendix A. 

The presented model has demonstrated high prediction accuracy and high computational efficiency using the closed-form solutions without resorting to FEM or any iteration-based calculations. Those advantages allow the presented model to be used for temperature prediction for large-scale parts and process-parameter planning through inverse analysis [45,46]. In the future, the presented model should be further developed for temperature prediction of multi-track and multi-layers scans. The applicability of the developed model on other widely used powder materials should be investigated. The intensity profile of the laser heat source should also be considered as a further improvement. The commonly observed defects of residual stress, porosity, and part deviation can be further investigated based on the calculated temperatures and temperature gradient because those defects were caused by the elevated temperature levels and large temperature gradients.

## 4. Conclusions

This work presented an analytical model for temperature prediction in powder bed metal additive manufacturing (PBMAM), also known as powder bed fusion (PBF) or selective laser melting (SLM). This model was developed based on physics with consideration of heat conduction, convection, and radiation, heat absorption, and latent heat affect. It was constructed from two closed-form solutions, namely the moving point heat source solution and the heat sink solution. The original heat sink solution was developed based on the heat transfer equations for convection and radiation. The influence of the chosen number of heat sinks on the predicted temperature profiles, temperature spatial gradient, and temperature-affected material property variation were investigated. An optimal number of heat sinks was determined by calibration with documented experimental values. The presented model was validated against documented experimental values under different process conditions. Close agreements were observed upon validation. 

The presented model has promising short computational time without resorting to FEM or any iteration-based simulations, which was confirmed from the recorded computational time. With the heat sink solution, the complete understanding of the heat transfer mechanism in PBMAM can be implemented effectively and efficiently. The employed heat sink solution improves the prediction accuracy, and thus the usefulness of the analytical modeling in real applications, specifically the temperature investigation in PBMAM. It can be used for temperature prediction of large-scale parts and process-parameter planning through inverse analysis because of the high computational efficiency. The calculated temperature and material property variation allows further investigations of residual stress, porosity, and part deviations, which are caused by repeated rapid heating and solidification. 

## Figures and Tables

**Figure 1 materials-12-02568-f001:**
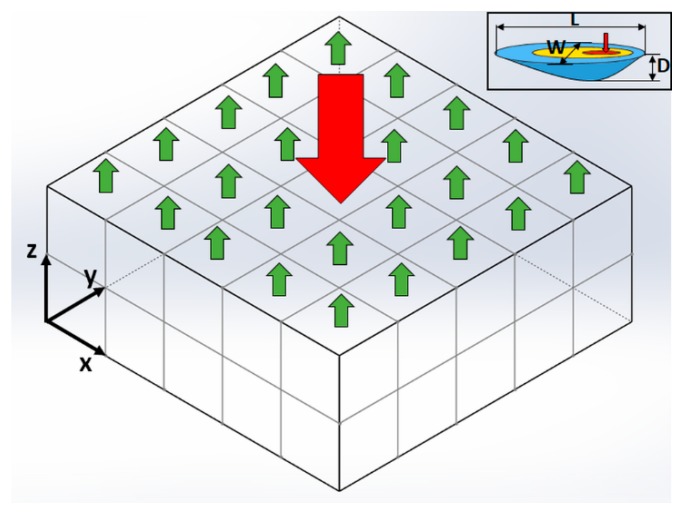
Schematic drawing of the heat transfer mechanism in PBMAM. The red arrow represents heat input from the laser source. The green arrows represent the heat loss from part boundary due to convection and radiation. Here, *x*, *y*, and *z* denote coordinate directions, while *L*, *W*, and *D* denote the molten pool length, width, and depth, respectively.

**Figure 2 materials-12-02568-f002:**
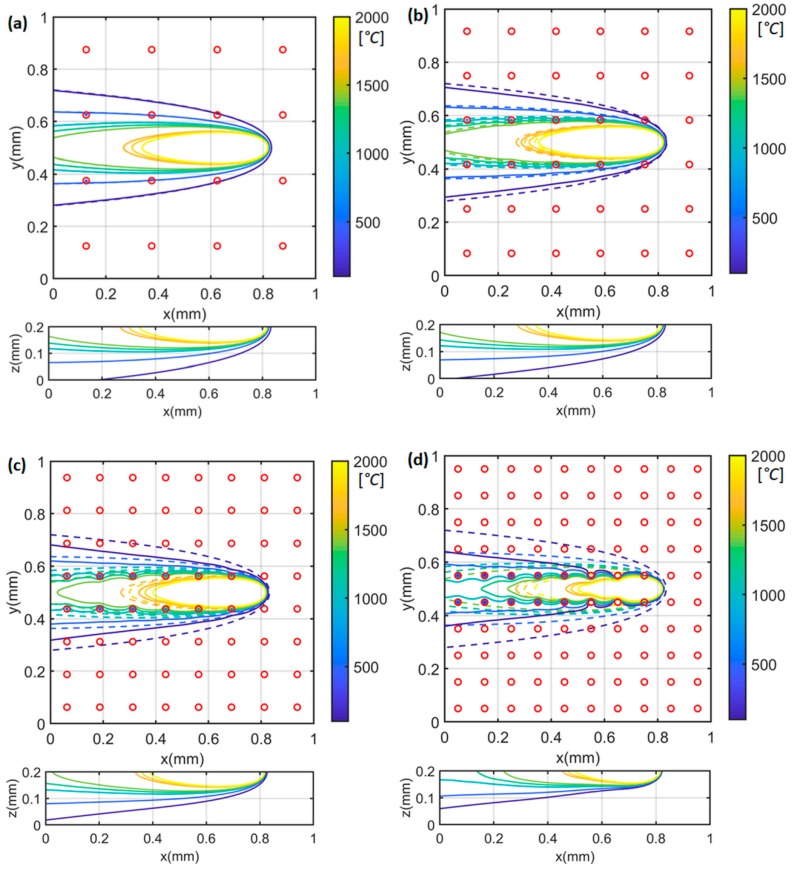
Calculated three-dimensional temperature distribution with (**a**) 4 × 4 = 16 heat sinks, (**b**) 6 × 6 = 36 heat sinks, (**c**) 8 × 8 = 64 heat sinks, and (**d**) 10 × 10 = 100 heat sinks. The dash lines represent the calculated temperature profiles without consideration of heat loss (zero heat sink). The solid lines represent the calculated temperature profiles with consideration of heat loss. Red circles represent the centers of heat sinks. The heat source is located at x = 0.8 mm, y = 0.5 mm.

**Figure 3 materials-12-02568-f003:**
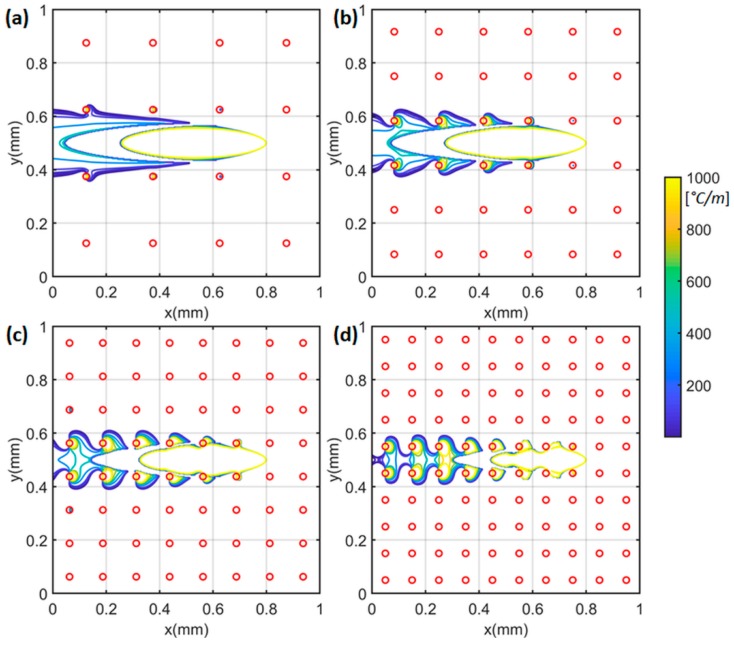
Calculated temperature gradient with (**a**) 4 × 4 = 16 heat sinks, (**b**) 6 × 6 = 36 heat sinks, (**c**) 8 × 8 = 64 heat sinks, and (**d**) 10 × 10 = 100 heat sinks. The moving laser is located at x = 0.8 mm, y = 0.5 mm. Red circles represent the centers of heat sinks.

**Figure 4 materials-12-02568-f004:**
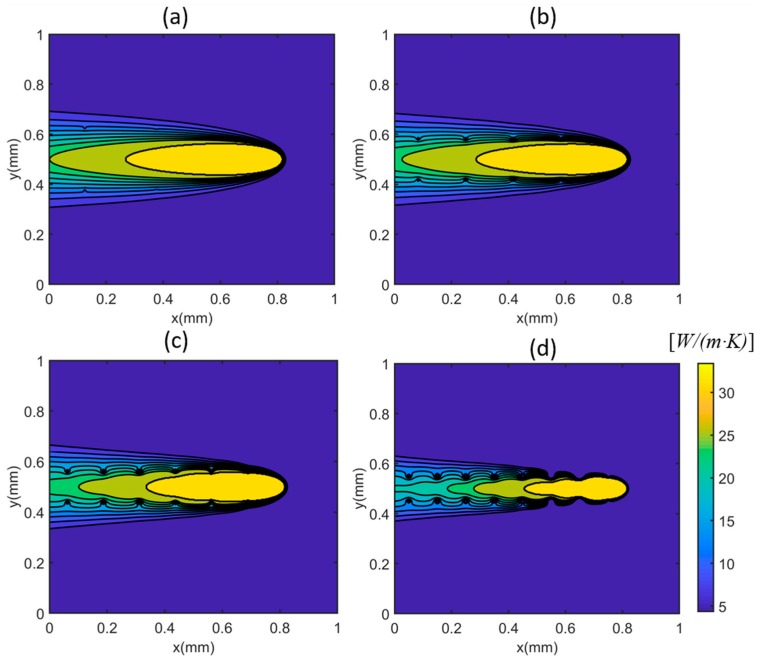
Calculated variations on thermal conductivity due to the single-track scans with different numbers of heat sinks: (**a**) 4 × 4 = 16 heat sinks, (**b**) 6 × 6 = 36 heat sinks, (**c**) 8 × 8 = 64 heat sinks, and (**d**) 10 × 10 = 100. The moving laser is located at x = 0.8 mm, y = 0.5 mm.

**Figure 5 materials-12-02568-f005:**
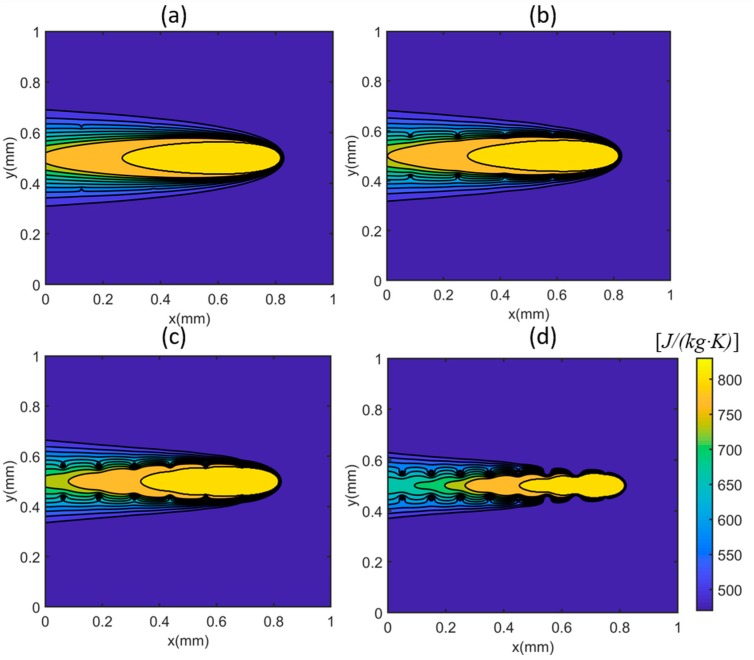
Calculated variations on specific heat due to the single-track scans with different numbers of heat sinks: (**a**) 4 × 4 = 16 heat sinks, (**b**) 6 × 6 = 36 heat sinks, (**c**) 8 × 8 = 64 heat sinks, and (**d**) 10 × 10 = 100. The moving laser is located at x = 0.8 mm, y = 0.5 mm.

**Figure 6 materials-12-02568-f006:**
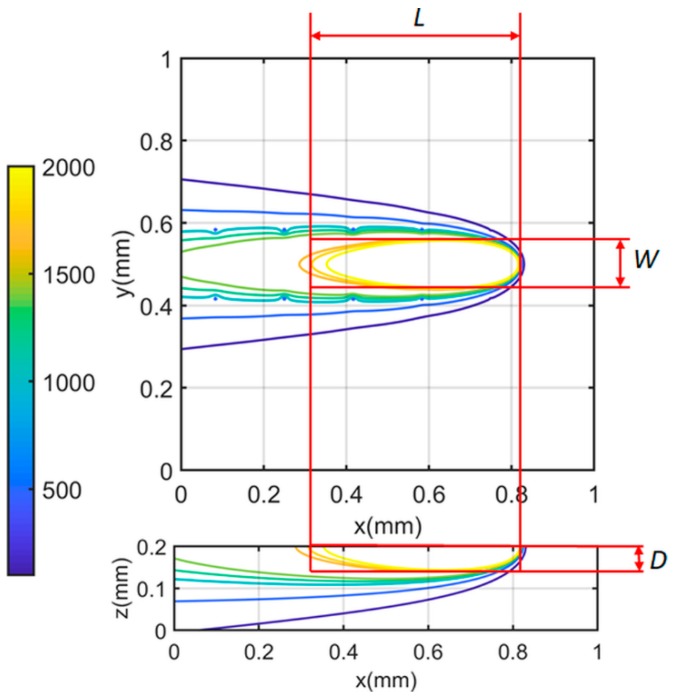
Calculation of molten pool dimensions from a three-dimensional temperature profile. *L*, *W*, and *D* represent the molten pool length, molten pool width, and molten pool depth, respectively.

**Figure 7 materials-12-02568-f007:**
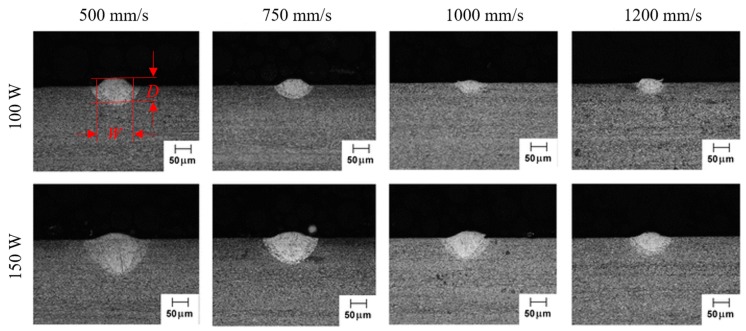
Experimental measurements of molten pool dimensions based on the solidification microstructure [2]. *W* and *D* denotes molten pool width and molten pool depth, respectively.

**Figure 8 materials-12-02568-f008:**
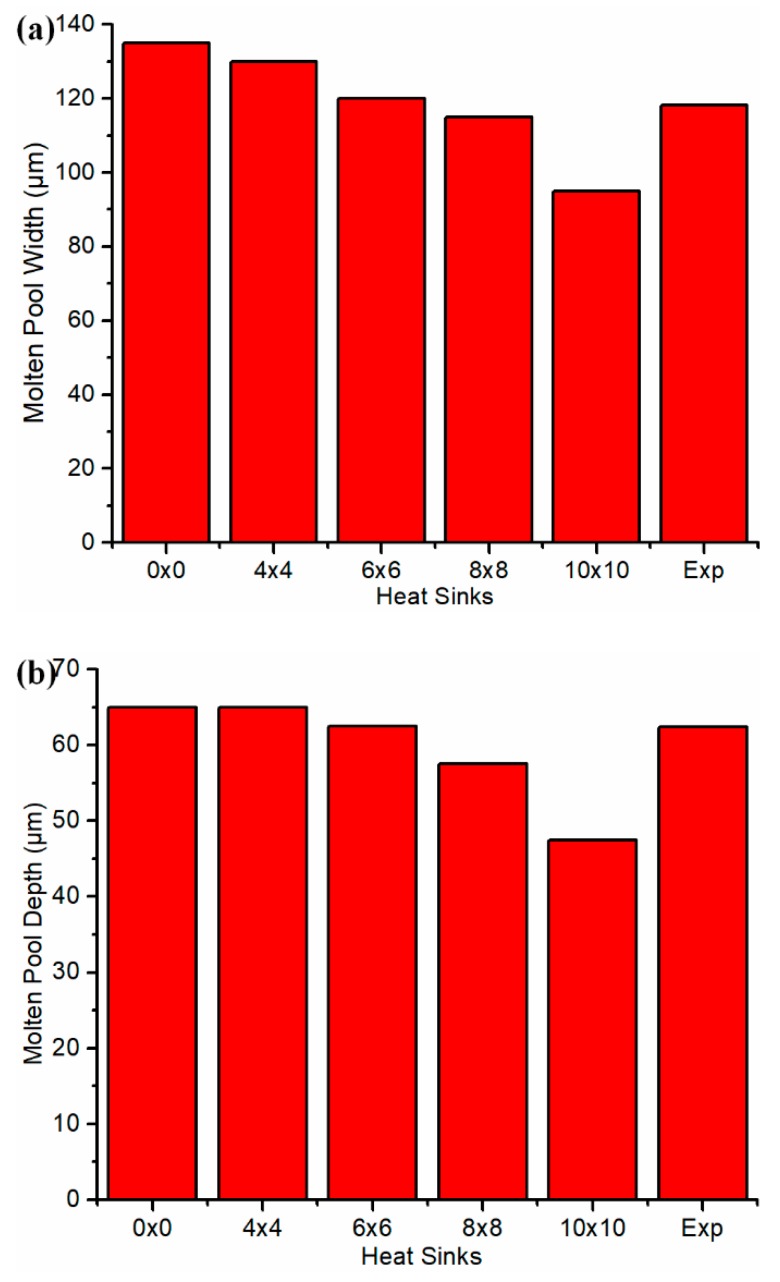
Determination of the number of heat sinks from the comparison between calculated molten pool dimensions and experimental measurements. Note: Exp denotes documented experimental values under test 1 conditions. (**a**) molten pool width (**b**) molten pool depth.

**Figure 9 materials-12-02568-f009:**
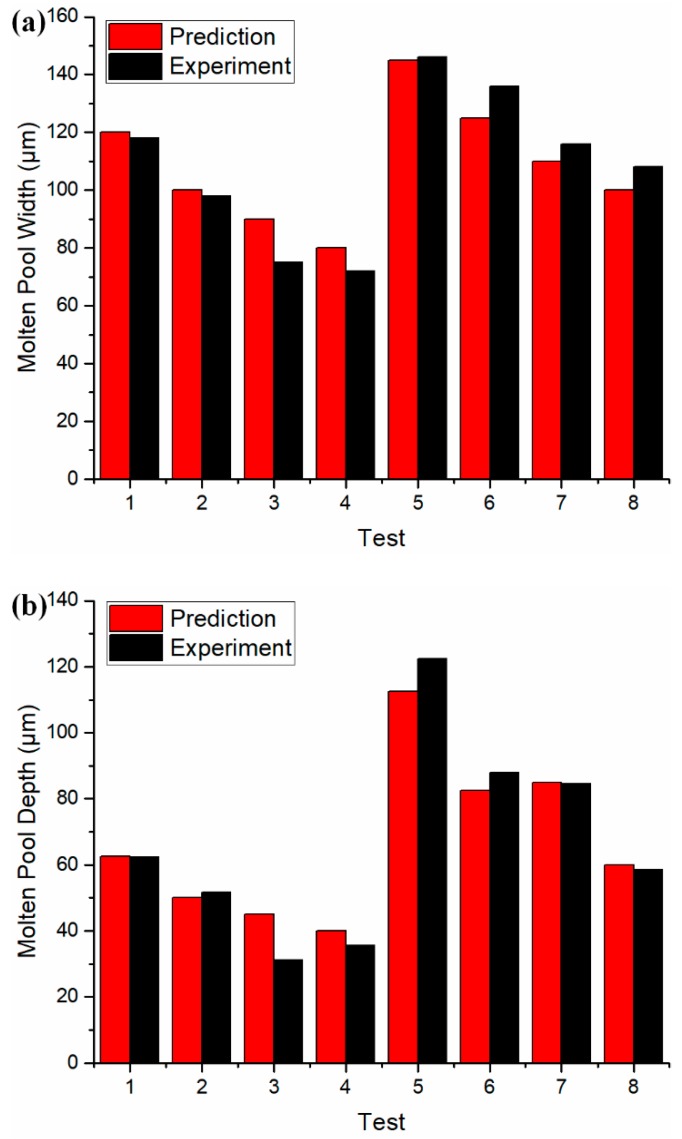
Validation of calculated molten pool dimensions to the experimental measurements under different process conditions. (**a**) Molten pool width, (**b**) molten pool depth.

**Table 1 materials-12-02568-t001:** Materials properties and thermophysical properties of Ti6Al4V [12,42,43,44].

Name	Symbol	Value	Unit
Density	ρ	4428	kg/m^3^
Thermal conductivity (powder at $T_{0}$)	$k_{p}$	6.6	W/(m∙K)
Thermal conductivity (solid)	$k_{s}$	$-0.797+18.2\times 10^{-3}T-2\times 10^{-6}\;T^{2}$ (T<1923 K) 33.4 (*T* > 1923 *K*)	W/(m∙K)
Specific heat (powder at $T_{0}$)	$c_{p}$	580	J/(kg∙K)
Specific heat (solid)	$c_{s}$	$411.5+2\times 10^{-1}T-5\times 10^{-7}T^{2}$ (T<1923 K) 830 (*T* > 1923 *K*)	J/(kg∙K)
Latent heat	$H_{f}$	365,000	J/kg
Room temperature	$T_{0}$	20	°C
Solidus temperature	$T_{s}$	1605	°C
Liquidus temperature	$T_{l}$	1655	°C
Heat convection coefficient	h	24	W/(m^2^∙K)
Emissivity	ε	0.9	1
Stefan-Boltzmann constant	σ	5.67 $\times \;10^{-8}$	W/(m^2^∙K^4^)

Note: Solid thermal conductivity and solid specific heat are only used in the investigation of materials’ property variations.

**Table 2 materials-12-02568-t002:** Process parameters of PBMAM of Ti6Al4V.

Name	Symbol	Value	Unit
Laser Powder	P	100, 150	W
Absorption	η	0.77	1
Scanning Velocity	V	500, 750, 1000, 1200	mm/s

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
