# Peer review of "Analytical Thermal Modeling of Metal Additive Manufacturing by Heat Sink Solution"

_materials, 2019, doi:10.3390/ma12162568_

Round 1

Reviewer 1 Report

The study presents an analytical approach to the modeling of heat transfer in laser-based metal additive manufacturing with a significant reduction of computational costs, as compared to FEM approaches. The results appear to strongly confirm the validity of the presented model. Nevertheless I have some comments: 

Equation (1) is a kind of continuity equation and apparently well-known, at least amongst heat transfer specialists. In order to adress a broader readership, I would recommend to add some more references concerning the origin of the heat balance equation and the derivation of the heat conduction equation from it.   

The presentation of equation (3) occurs together with the introduction of some parameters, namely R and Xi, which do not appear in eq. (3), but later in eq. 4. The power P is not being introduced at all. The authors should correct this.

It is not obvious, where the theory known from literature ends and where the new model starts. As this is mainly a theoretical paper, in my opinion the presentation of the model could do with a few more formulas explaining the process of setting up the theoretical model. This would make it more accessible for rather experimental scientists.

Fig. 9: The experimental results appear to be in good agreement with the prediction. Is there any error analysis of the experimental values? How many experiments were performed per parameter set?  

Author Response

The authors appreciate the reviewer’s efforts on improving our manuscript. A detailed response to the reviewer’s comments is attached.

Reviewer 2 Report

The authors provides analytical thermal modeling for metal additive manufacturing.

Heat sink solution was utilized to consider the heat transfer boundary condition.

The developed method can reduce computational time drastically compared to the conventional FEM calculation.

However, validation of this method is done only by reproducing the molten pool width and depth.

That means the method does not based on physics.  It may be sufficient from view point of engineering, but I cannot rely on this method as science.

Author Response

(The authors gave the same response as above.)

Reviewer 3 Report

In the paper entitled “Analytical Thermal Modeling of Metal Additive Manufacturing by Heat Sink Solution”, Authors presents an explicit and closed-form solution, namely heat sink solution, to consider the heat transfer boundary condition. Presented paper is characterized by good scientific level.

I leave my objections to the author's reflection:

1) For example, the statement "More information can be found in the review literature" is inappropriate. Please correct.

Author Response

(The authors gave the same response as above.)

Reviewer 4 Report

In the paper an analytical model for temperature prediction in powder bed metal additive manufacturing is presented. The analytical formulation is presented in the first part of the paper: the model is finally validated experimentally.

The subject of the paper is interesting and in the field of “Materials” journal. The paper is well written and the proposed model is in agreement with experimental results.

I would only suggest the Authors to consider the following comments, as possible points of further discussion in the manuscript

Abstract

The abstract is 220 words and exceeds the maximum allowed length (200 words). Please revise the abstract in order to reduce the number of words to 200 words.

Materials and Methods The Authors should more clearly explain what is the “heat sink solution” (especially for non-initiated readers).

Results and Discussion Page 5, lines 207-210: “The number of heat sinks was determined by calibration based on the experimental measurement of molten pool dimensions in the literature [2]. The documented molten pool width and depth were measured based on the solidification microstructure using an optical microscope.” The model is calibrated by considering literature measurements of the molten pool dimension. Does an error on the molten pool dimension affect the results? Is the model sensible to small variations of the molten pool dimension? Please comment on this.

Page 10 Figure 15, lines 239-240: 6 x 6 heat sinks permit to obtain results closest to the experimental results. Is the optimal number of heat sinks found in this paper material dependent? Is there a way to assess the optimal number of heat sinks analytically or is an experimental calibration always necessary?

Very minor comments

Page 6 line 221: please replace “infinity” with “infinite”.

Author Response

(The authors gave the same response as above.)

Round 2

Reviewer 2 Report

The paper can be accepted for publication of this journal.

The authors provide analytical thermal modeling for metal additive manufacturing. Heat sink solution was utilized to consider the heat transfer boundary condition. The developed method can reduce computational time drastically compared to the conventional FEM calculation.

In the revised version, the authors provides sufficient information about the foundation of physics for their method, including additional references of the former researches. The reader will have interests in this field and take valuable information from this manuscript.